# Genomic and Biological Characterization of a Novel *Proteus mirabilis* Phage with Anti-Biofilm Activity

**DOI:** 10.3390/v17111419

**Published:** 2025-10-25

**Authors:** Yan Liang, Nana Li, Shenghui Wan, Yanfang Li, Yuwan Li, Yonggang Qu

**Affiliations:** 1College of Animal Science and Technology, Shihezi University, Shihezi 832003, China; liangyan@shzu.edu.cn (Y.L.); linana1217@foxmail.com (N.L.); wanshenghui0807@foxmail.com (S.W.); yanfangli@shzu.edu.cn (Y.L.); 2College of Food and Bioengineering, Henan University of Science and Technology, Luoyang 471023, China

**Keywords:** *Proteus mirabilis*, bacteriophage, biological characteristics, whole genome sequencing, multidrug resistance

## Abstract

The emergence of multidrug-resistant (MDR) *Proteus mirabilis* poses a significant threat in porcine farming and public health, highlighting the need for alternative biocontrol agents. This study aimed to isolate and characterize a lytic bacteriophage with therapeutic potential against MDR *P. mirabilis*. Using the clinical MDR *P. mirabilis* strain Pm 07 as host, a bacteriophage, vB_Pmc_P-07 (P-07), was successfully isolated from fecal and sewage samples via an enrichment protocol. Phage P-07 forms plaques surrounded by a distinct translucent “halo,” suggesting the production of depolymerase. It achieved high titers of up to 1.40 × 10^8^ PFU/mL and exhibited a narrow host range, high stability across a broad range of temperatures (40–60 °C) and pH (4–12), as well as considerable anti-biofilm activity. An optimal multiplicity of infection (MOI) of 0.001 was determined. Whole-genome sequencing revealed a linear double-stranded DNA genome of 58,582 bp with a GC content of 46.91%, encoding 63 open reading frames. Crucially, no virulence or antibiotic resistance genes were detected, supporting its safety profile. Phylogenetic analysis classified P-07 within the *Casjensviridae* family, closely related to phages PM87 and pPM01. These findings indicate that phage P-07 is a novel, safe, and effective lytic phage with strong potential as a biocontrol agent against biofilm-forming MDR *P. mirabilis* in swine.

## 1. Introduction

*Proteus mirabilis* is a Gram-negative, rod-shaped bacterium widely distributed in natural environments [1]. As an opportunistic pathogen, it causes infections in the digestive, respiratory, and urinary tracts of both humans and animals, with rising incidence rates in recent years [2,3].

In swine farming, *P. mirabilis* represents a growing concern. While often presenting with subtle clinical manifestations, its ability to form biofilms and serve as a reservoir for multidrug resistance genes poses serious challenges to farm management and animal health [4,5]. Moreover, the zoonotic potential of this pathogen is underscored by its detection in retail meat products, including pork. Strains isolated from meat sources show remarkable genetic similarity and virulence gene overlap with human clinical isolates, suggesting a plausible foodborne transmission route [6,7].

The global spread of multidrug-resistant (MDR) bacteria, causing an estimated 700,000 deaths annually, has been recognized by the WHO as a critical public health threat. This urgency has renewed interest in bacteriophage therapy as a promising alternative to conventional antibiotics. Phages offer unique advantages including high specificity, self-replication capability, and a favorable safety profile [8,9]. Their potential in livestock production is increasingly acknowledged, with documented efficacy against pathogens like *Escherichia coli* and *Salmonella* in poultry and swine [10,11].

Nevertheless, specific research on phage therapy targeting *P. mirabilis* in agricultural settings remains limited. To address the knowledge gap, this study aimed to isolate and characterize a lytic bacteriophage specific to porcine-originated MDR *P. mirabilis*, investigating its biological characteristics, genomic features, and anti-biofilm potential to develop a foundation for phage-based control strategies in swine farming.

## 2. Materials and Methods

### 2.1. Bacterial Strains and Sample Collection

MDR *P. mirabilis* strain Pm 07 and other tested strains (41 strains of *P. mirabilis*, 18 strains of *E. coli*, 3 strains of *Klebsiella pneumoniae*, and two strains of *Staphylococcus aureus*) were isolated from Xinjiang Uygur Autonomous Region of China and identified by the Animal Infectious Disease Laboratory at Shihezi University, and their detailed information is provided in Appendix A. The three primary MDR strains investigated in this study—Pm 07, Pm 08, and Pm 13—were all confirmed as ESBL-producing *P. mirabilis*. Pm 07 exhibited resistance to 18 antibiotics, including cefalexin (CA), penicillin (PG), amoxicillin (AM), enrofloxacin (ENR), ciprofloxacin (CIP), norfloxacin (NF), trimethoprim-sulfamethoxazole (SXT), tetracycline (TC), doxycycline (DOX), gentamicin (GN), kanamycin (KAN), tobramycin (TOB), florfenicol (FFC), chloramphenicol (CL), polymyxin B (PB), erythromycin (ERY), vancomycin (VAN), and imipenem (IMP). Pm 08 was resistant to 19 antibiotics, including cefotaxime (CTX), cefalexin (CA), penicillin (PG), amoxicillin (AM), enrofloxacin (ENR), ciprofloxacin (CIP), norfloxacin (NF), trimethoprim-sulfamethoxazole (SXT), tetracycline (TC), doxycycline (DOX), gentamicin (GN), kanamycin (KAN), tobramycin (TOB), florfenicol (FFC), chloramphenicol (CL), polymyxin B (PB), erythromycin (ERY), vancomycin (VAN), and imipenem (IMP). Strain Pm 13 showed resistance to 17 antibiotics, including ceftazidime (CAZ), cefalexin (CA), penicillin (PG), amoxicillin (AM), enrofloxacin (ENR), ciprofloxacin (CIP), norfloxacin (NF), trimethoprim-sulfamethoxazole (SXT), tetracycline (TC), doxycycline (DOX), kanamycin (KAN), tobramycin (TOB), florfenicol (FFC), chloramphenicol (CL), polymyxin B (PB), erythromycin (ERY), vancomycin (VAN), and imipenem (IMP). All antibiotic susceptibility testing was performed by the Kirby-Bauer disk diffusion method on Mueller-Hinton agar, with results interpreted according to the Clinical and Laboratory Standards Institute (CLSI) guidelines. In addition, the crystal violet staining assay confirmed varying biofilm-forming capacities among the isolates: strain Pm 07 exhibited strong biofilm formation with an OD_590_ value > 4 × the negative control (DNC), strain Pm 08 demonstrated moderate capacity (2× DNC < OD_590_ ≤ 4× DNC), and strain Pm 13 showed weak capacity (DNC < OD_590_ ≤ 2× DNC). (data not yet published).

Feces and wastewater were collected from large-scale pig farms in the Shihezi region of Xinjiang and underwent subsequent phage isolation.

### 2.2. Enrichment of Bacteriophages

Bacteriophage enrichment was performed using a modified protocol based on the method by Jamalludeen et al. [12]. Briefly, SM buffer (50 mM Tris-HCl [pH 7.5], 100 mM NaCl, 8 mM MgSO_4_, and 0.01% gelatin) was added to mixed fecal-sewage samples, followed by static sedimentation at 4 °C for 24–48 h. The supernatant was then collected via centrifugation (2910× *g*, 10 min, 4 °C) and filtered through 0.45 μm membranes for storage at 4 °C. Concurrently, Pm 07 was inoculated into LB broth and cultured with shaking until the logarithmic growth phase was reached (37 °C). For phage amplification, 40 mL of filtered fecal sample was mixed with 200 μL of fresh Pm 07 culture in an equal volume of 2× LB broth, and the mixture was incubated with shaking (37 °C, 180 r/min) for 12–16 h. Finally, the culture was centrifuged (16,760× *g*, 15 min, 4 °C), and the supernatant was passed through a 0.22 μm sterile membrane to remove bacterial debris and obtain the purified bacteriophage stock solution.

### 2.3. Isolation and Purification of Bacteriophage

The presence of lytic bacteriophages in the crude phage solution was detected using the spot assay [13]. Briefly, 200 μL log period *P. mirabilis* culture was spread evenly onto LB solid medium using a sterile cotton swab. After the bacterial suspension was absorbed, 10 μL of the crude phage solution was spotted onto the plate. The plates were air-dried completely, inverted, and incubated at 37 °C for 12–16 h to observe plaque formation.

Phage purification was subsequently carried out using the double-layer agar method [14]. Individual plaques were aseptically picked and transferred into SM buffer, followed by incubation overnight at 4 °C. The suspension was then centrifuged at 11,640× *g* for 10 min and sterilized by filtration through a 0.22 μm membrane. For phage amplification, 100 μL of host bacterial suspension and 100 μL of phage solution were mixed in LB broth and incubated with shaking (180 r/min, 37 °C) for 6–8 h. The culture was centrifuged again (11,640× *g*, 10 min), and the supernatant was collected. Ten-fold serial dilutions of the supernatant were prepared. From each dilution, 100 µL was mixed thoroughly with 100 µL of host bacteria, allowed to adsorb statically, mixed with molten soft agar, and poured onto LB agar plates. After solidification, the plates were incubated at 37 °C. This purification procedure was repeated 3 to 5 times until plaques with uniform morphology and size were obtained.

### 2.4. Concentration and Electron Microscopy Observation of Bacteriophage

Bacteriophage particles were concentrated using the polyethylene glycol (PEG8000) precipitation method [15]. Briefly, a 200 mL phage lysate was initially prepared. The supernatant was collected by centrifugation (7450× *g*, 15 min, 4 °C) and sterilized via membrane filtration. To remove residual host nucleic acids, DNase I and RNase A were added to the filtrate at a final concentration of 1 μg/mL each, followed by incubation at 37 °C for 1 h. Subsequently, NaCl was added to a final concentration of 1 M, and the mixture was incubated on ice for 2 h. After centrifugation (16,760× *g*, 20 min, 4 °C), the pellet was discarded. PEG8000 was added to the supernatant to a final concentration of 10% (*w*/*v*), dissolved completely by gentle stirring, and the solution was incubated on ice for another hour. The mixture was then centrifuged (11,640× *g*, 10 min, 4 °C) again. The resulting pellet was resuspended in 1 mL of SM buffer and incubated at 4 °C for 1 h. An equal volume of chloroform was added for extraction. Following centrifugation (1050× *g*) at 4 °C for 15 min, the aqueous phase was carefully collected as the concentrated phage preparation.

For electron microscopy, the concentrated phage solution (50 µL) was applied onto copper grids (with carbon support film) and allowed to adsorb for 15 min. Negative staining was performed using 2% (*w*/*v*) phosphotungstic acid for 5 min. Excess stain was removed with filter paper, and the grids were irradiated with infrared light for 30 min. Samples were visualized using an HT7700 transmission electron microscope (Hitachi High-Technologies Corporation, Tokyo, Japan).

### 2.5. Lytic Spectrum Profiling

The host range of the isolated bacteriophage was determined against a panel of bacterial strains, including 41 *P. mirabilis*, 18 *E. coli*, 3 *K. pneumoniae*, and 2 *S. aureus* isolates (excluding the host strain), using a spot assay. Briefly, 200 μL of fresh logarithmic-phase bacterial culture was spread evenly onto LB agar plates. After the surface dried, 5 μL of phage suspension was spotted onto pre-designated areas. The plates were then incubated overnight at 37 °C. A clear lytic zone observed at the spot location, contrasted with confluent bacterial growth in the surrounding control area, was interpreted as a positive lytic activity against the corresponding strain.

### 2.6. Optimal Multiplicity of Infection (MOI) Determination

The optimal multiplicity of infection (MOI) was determined as previously described [16,17]. Briefly, mixtures containing equal volumes of host bacterial suspension and phage solutions at different MOIs (0.00001, 0.0001, 0.001, 0.01, 0.1, 1, 10, 100) were prepared in 800 μL of LB broth and mixed thoroughly. After 4 h of incubation with shaking (180 r/min, 37 °C), the cultures were centrifuged (11,640× *g*, 10 min, 4 °C). The supernatants were filtered through 0.22 μm membranes, and phage titers were determined using the double-layer agar method. All assays were performed in triplicate, and the results are presented as the mean values.

### 2.7. One-Step Growth Curve Analysis

The one-step growth curve of the phage was analyzed according to the method described by Pajunen et al. [18]. Briefly, the phage and its host bacteria were mixed at the optimal multiplicity of infection (MOI) and incubated at 37 °C for 10 min to allow adsorption. The mixture was then centrifuged (16,760× *g*, 10 min) to remove unadsorbed phages. The pellet was washed twice with phosphate-buffered saline (PBS), resuspended in 50 mL of fresh LB broth, and incubated at 37 °C with shaking (180 r/min). Samples (1 mL each) were collected at predetermined time points over a period of 180 min, immediately centrifuged (16,760× *g*, 2 min), and the supernatant was subjected to phage titration using the double-layer agar plaque assay. All titrations were performed in triplicate, and the phage titers are expressed as the mean values.

### 2.8. Thermal Stability Assessment

The thermal stability of the phage was evaluated by incubating 2 mL aliquots of the phage suspension in water baths at temperatures ranging from 40 °C to 80 °C (40 °C, 50 °C, 60 °C, 70 °C, 80 °C). At 20, 40, and 60 min intervals, 200 μL samples were withdrawn and immediately centrifuged (16,760× *g*, 5 min, 4 °C) to remove any debris. The phage titer in the supernatant was subsequently determined using the double-layer agar plaque assay [19]. The experiment was performed in triplicate, and the results are expressed as the mean ± standard deviation.

### 2.9. pH Tolerance Evaluation

The pH tolerance of the phage was assessed by exposing the viral particles to a broad range of pH conditions. Aliquots (100 μL) of the phage suspension were mixed with 900 μL of LB broth pre-adjusted to different pH values (ranging from 1 to 14). The mixtures were incubated statically at 37 °C for 2 h. Subsequently, the samples were centrifuged (16,760× *g*, 5 min, 4 °C) to pellet any potential debris or aggregates, and the surviving phage titer in the supernatant was determined using the double-layer agar plaque assay. The experiment was conducted in triplicate, and the results are presented as the mean ± standard deviation.

### 2.10. Bacteriophage-Mediated Anti-Biofilm Activity Analysis

The anti-biofilm activity of the bacteriophage against *P. mirabilis* was evaluated using a crystal violet staining assay. Prior to the phage treatment, the biofilm formation dynamics of the host strain *P. mirabilis* 07 were characterized. The bacteria were inoculated and cultured as described below, and the biofilm biomass was quantified at 4, 8, 12, 24, 36, and 48 h to establish the growth kinetics. This preliminary study confirmed that a mature biofilm was formed at 24 h, which was consequently selected as the time point for phage application in the subsequent anti-biofilm assay.

For the main assay, host bacteria were inoculated (1:100) into fresh liquid medium and incubated at 37 °C with shaking at 180 r/min until the OD_600_ reached the logarithmic phase. A 1 mL aliquot of the culture was transferred to each well of a 24-well plate and incubated statically at 37 °C for 24 h to allow for the development of mature biofilms. After careful removal of the planktonic culture, the wells were gently rinsed three times with PBS to remove non-adherent cells. The established biofilms were then treated with 1 mL of phage suspensions at concentrations of 5.6 × 10^8^ PFU/mL or 5.6 × 10^7^ PFU/mL, using LB broth as a negative control. Following static incubation at 37 °C for 2, 4, and 6 h post-phage addition, the wells were washed again three times with PBS. The remaining adherent biofilm biomass was quantified by crystal violet staining, and the absorbance was measured at 590 nm. Results are shown as the mean ± SD.

### 2.11. Genomic DNA Extraction, Sequencing, and Bioinformatics Analysis

Bacteriophage genomic DNA was extracted using the standard phenol-chloroform method. Briefly, a purified high-titer phage lysate was treated with DNase I and RNase A (1 µg/mL each) at 37 °C for 30 min to remove residual host nucleic acids. Subsequently, EDTA (pH 8.0), Proteinase K, and SDS were added to final concentrations of 20 mM, 50 µg/mL, and 0.5% (*w*/*v*), respectively, followed by incubation at 56 °C for 1 h to digest the capsid proteins. Nucleic acids were extracted with an equal volume of phenol-chloroform-isoamyl alcohol (25:24:1) and precipitated with isopropanol. The DNA pellet was washed with 75% ethanol, air-dried, and finally resuspended in nuclease-free water.

The purified DNA underwent Illumina sequencing at Huitong Biotechnology Co., Ltd., Hangzhou (Zhejiang, China). Genome assembly employed Newbler software (Version:2.9). Protein-coding genes were predicted using GeneMarkS. Antibiotic resistance genes were identified via CARD (http://card.mcmaster.ca/, accessed on 16 February 2025), while virulence factors were detected using the Center for Genomic Epidemiology platform (http://www.genomicepidemiology.org/services/, accessed on 16 February 2025). Circular genome maps were generated with CGView Server (https://js.cgview.ca/, accessed on 16 February 2025).

The terminase large subunit—a signature phage gene critical for phylogenetic analysis [20]—underwent BLAST (https://blast.ncbi.nlm.nih.gov/Blast.cgi, accessed on 16 February 2025) alignment on NCBI. Phylogenetic trees were constructed using MEGA 11.0.

The observation of translucent halos surrounding the plaques of phage P-07 suggested the potential production of depolymerase, which was subsequently investigated through genomic analysis. Following gene annotation, phylogenetic analysis of the phage tail fiber protein was performed using MEGA 11.0 to elucidate its evolutionary relationships. Furthermore, the tertiary structure of the tail fiber protein was predicted using the Phyre2 online server (http://www.sbg.bio.ic.ac.uk/phyre2, accessed on 9 September 2025).

### 2.12. Data Analysis

All statistical analyses utilized GraphPad Prism 9.0 (GraphPad Software, Inc., San Diego, CA, USA). Significance was determined by multiple *t*-tests. Differences were considered statistically significant at *p* < 0.05 and highly significant at *p* < 0.01. Error bars denote the standard deviation (SD) of means.

## 3. Results

### 3.1. Isolation, Purification, and Electron Microscopy Morphology

A lytic bacteriophage infecting *Proteus mirabilis* was isolated from fecal-wastewater samples and designated vB_Pmc_P-07 (abbreviated as P-07) in accordance with the guidelines of the International Committee on Taxonomy of Viruses (ICTV). Plaque assay showed that P-07 formed clear plaques surrounded by distinct translucent halos (Figure 1A). Transmission electron microscopy (TEM) imaging at 80 kV revealed that P-07 has an icosahedral head and a short tail, with approximate dimensions of 60 nm in head diameter and 12 nm in tail length (Figure 1B).

### 3.2. Lytic Spectrum Determination

Bacteriophage P-07 exhibited strict specificity, lysing only *P*. *mirabilis* and displaying no cross-species lytic activity against *E*. *coli*, *K*. *pneumoniae*, or *S*. *aureus*. When tested against a panel of 42 *P. mirabilis* isolates (including the host strain), the lysis rate was 21.42% (see Appendix A).

### 3.3. Optimal Multiplicity of Infection (MOI) and One-Step Growth Curve of P-07

The optimal multiplicity of infection (MOI) for phage P-07 was determined to be 0.001, yielding a peak titer of 9.66 × 10^9^ PFU/mL (Figure 2A). The one-step growth curve revealed a latent period of approximately 30 min, followed by a burst phase. The phage titer increased rapidly during the exponential rise period (30–60 min) and then more gradually until reaching a plateau at 100 min post-infection. The final burst size was calculated to be 474.99 PFU/cell, with a plateau titer of 8.830 log PFU/mL (Figure 2B).

### 3.4. Thermal and pH Stability of Phage P-07

Phage P-07 demonstrated notable thermal stability, maintaining stable titers at temperatures ranging from 40 to 60 °C for up to 40 min, with only a marginal decrease observed after 60 min of incubation (Figure 3A). However, at 70 °C, the phage titer declined rapidly after 20 min and was completely inactivated by 40 min. At 80 °C, the phage was completely inactivated after 20 min of incubation, with no plaques detected.

The phage also exhibited high tolerance to a broad pH range. High activity was retained within pH 5 to 11, while complete inactivation occurred under strongly acidic (pH ≤ 4) or strongly alkaline (pH ≥ 12) conditions (Figure 3B). Collectively, these findings indicate that P-07 possesses robust stability under various environmental conditions.

### 3.5. Biofilm Clearance Efficacy Against P. Mirabilis 07

*P. mirabilis* 07 exhibited a typical biofilm growth dynamic, characterized by a steady increase in biomass from 4 to 24 h, peaking at 24 h. Subsequently, the biofilm entered a decline phase, resulting in a reduction in total biomass (Appendix A).

Bacteriophage P-07 demonstrated significant anti-biofilm activity against *P. mirabilis* 07 in a concentration and time dependent manner. After 2 h of treatment, both low-concentration (0.603 ± 0.031) and high-concentration (0.533 ± 0.022) groups showed significantly reduced OD_590_ values compared to the control group (0.814 ± 0.063) (*p* < 0.05). This inhibitory effect was maintained at 4 h, with OD_590_ values remaining significantly lower in both treatment groups (low-concentration: 0.564 ± 0.040; high-concentration: 0.518 ± 0.029) versus control (0.823 ± 0.131) (*p* < 0.05). By 6 h, the anti-biofilm effect became more pronounced, with both concentration groups (low-concentration: 0.469 ± 0.017; high-concentration: 0.436 ± 0.015) showing a highly significant reduction compared to the control (0.805 ± 0.094) (*p* < 0.01) (Figure 4A).

### 3.6. Strain-Dependent Anti-Biofilm Activity

Bacteriophage P-07 exhibited distinct anti-biofilm efficacy against preformed biofilms of *P. mirabilis* strains with varying biofilm-forming capacities. When treating 24 h mature biofilms formed by the moderate biofilm-forming strain *P. mirabilis* 08, the high-concentration P-07 significantly reduced OD_590_ values (0.603 ± 0.012) after 2 h compared to control (0.765 ± 0.043) (*p* < 0.05). At 4 h post-treatment, both low-concentration (0.573 ± 0.020) and high-concentration (0.529 ± 0.019) groups showed significant inhibition versus control (0.753 ± 0.032) (*p* < 0.05). After 6 h, while the low-concentration group (0.552 ± 0.016) demonstrated a significant reduction (*p* < 0.05), the high-concentration group (0.477 ± 0.015) showed highly significant efficacy compared to control (0.749 ± 0.016) (*p* < 0.01, Figure 4B).

For 24 h, mature biofilms formed by the weak biofilm-forming strain *P. mirabilis* 13; significant anti-biofilm effects were only observed after 6 h of treatment, where the high-concentration group (0.535 ± 0.011) exhibited significantly lower OD_590_ values than the control (0.662 ± 0.012) (*p* < 0.05, Figure 4C).

### 3.7. Differential Anti-Biofilm Efficacy Across Strains with Varying Biofilm-Forming Capacities

The anti-biofilm efficacy of P-07 demonstrated clear stratification corresponding to the biofilm-forming capacity of the bacterial strains. At low concentration, 2 h treatment reduced OD_590_ values by 0.210, 0.076, and 0.035 in 24 h mature biofilms formed by *P. mirabilis* 07 (strong biofilm-forming), *P. mirabilis* 08 (moderate biofilm-forming), and *P. mirabilis* 13 (weak biofilm-forming), respectively, compared to their controls. These reductions increased in a time-dependent manner, reaching 0.258, 0.180, and 0.054 after 4 h, and 0.336, 0.197, and 0.080 at 6 h. Throughout all time points, low-concentration P-07 showed significantly stronger clearance efficacy against mature biofilms of *P. mirabilis* 07 compared to those of both *P. mirabilis* 08 and *P. mirabilis* 13 (*p* < 0.05).

The high-concentration group demonstrated enhanced biofilm clearance against mature biofilms: 2 h treatment decreased OD_590_ values by 0.280, 0.161, and 0.051 for 24 h biofilms of *P. mirabilis* 07, 08, and 13, respectively. The reductions increased to 0.304, 0.225, and 0.090 at 4 h, and reached 0.367, 0.272, and 0.128 at 6 h. Similarly, high-concentration P-07 consistently exhibited significantly superior clearance efficacy against mature biofilms of *P. mirabilis* 07 compared to those of both *P. mirabilis* 08 and *P. mirabilis* 13 at all time points (*p* < 0.05).

### 3.8. Whole-Genome Analysis of Phage P-07

Whole-genome sequencing of phage P-07 yielded a complete double-stranded DNA genome of 58,582 bp with a GC content of 46.91%. Nucleotide composition analysis revealed the following distribution: 25.09% adenine (A), 28.01% thymine (T), 24.55% cytosine (C), and 22.36% guanine (G). The genome sequence has been deposited in GenBank under accession number PV817856.

Gene prediction and annotation identified 63 open reading frames (ORFs), including 15 transcribed in the forward direction and 48 in the reverse direction. Functional categorization revealed 30 ORFs encoding putative proteins, which were classified into four distinct modules: structural/packaging (orange), DNA replication/regulation (blue), auxiliary metabolic (green), and lysis (red). Notable functional assignments included: ORF24 (endolysin) in the lysis module; ORF46 (head maturation protease) in the metabolic module; ORF53 (DNA polymerase), ORF58 (primase), ORF2 (DNA modifier), ORF18 (methyltransferase), and ORF13/52/55 (exonucleases) in the DNA replication module; along with ORF25 (Rz), ORF39 (structural protein), ORF50/49 (terminase), ORF31 (tail fiber), ORF40 (tail terminator), ORF42 (head closure protein), ORF44 (capsid), ORF45 (decoration protein), ORF48 (head-tail connector), and ORF32-36/38 (tail tube proteins) in the structural/packaging module (Figure 5, Table 1).

Screening for antibiotic resistance genes and virulence factors using the CARD database and the Center for Genomic Epidemiology platform, respectively, revealed no identifiable resistance genes or virulence factors in the P-07 genome.

Phylogenetic analysis based on the terminase large subunit sequence positioned P-07 within the class *Caudoviricetes*, family *Casjensviridae*, and genus *Lavrentievirus*. The terminase large subunit showed 88% amino acid identity to those of *P. mirabilis* phages PM87 and pPM01, confirming its close evolutionary relationship with these known phages (Figure 6).

In-depth bioinformatic analysis was performed on ORF31, annotated as a tail fiber protein. Its tertiary structure was predicted using the Phyre2 online server. The prediction revealed two key functional domains: a C-terminal region (residues 178–225) exhibiting limited sequence coverage (19%) with lyase enzyme templates (c6p1yB_, c7a8yD_), and an N-terminal region (residues 22–115) showing significant structural homology (37% coverage) to the baseplate-tail complex of a marine siphophage (template c8gtcS_). The model dimensions for this baseplate-binding region are 38.817 Å (X) × 37.000 Å (Y) × 30.111 Å (Z) (Figure 7). Secondary structure composition analysis indicated a predominance of β-sheets (33%) over α-helices (9%), consistent with typical structural protein features (Figure 8). These findings suggest ORF31 is a multifunctional protein, with its N-terminal domain facilitating phage adsorption through baseplate integration and its C-terminal domain potentially enabling enzymatic degradation of host polysaccharides.

Building upon these structural predictions, we propose that ORF31 represents a “tail fiber-depolymerase fusion functional module,” a genomic feature commonly employed by phages to enhance biofilm penetration. This configuration ensures the synchronous transcription and assembly of the tail fiber with its associated enzymatic activity, thereby maximizing infection efficiency. The presence of an aspartate-rich domain within ORF31 is particularly significant, as specific aspartate residues have been experimentally demonstrated to serve as essential catalytic residues in polysaccharide-degrading enzymes; for instance, site-directed mutagenesis of key aspartates (Asp227 and Asp342) in an isomalto-dextranase completely abolished its hydrolytic activity [21]. This, coupled with the established role of aspartate in forming catalytically competent active sites through mechanisms such as hydrogen bonding and transition-state stabilization in glycoside hydrolases [22], provides strong genetic and mechanistic evidence that ORF31 encodes a depolymerase, equipping phage P-07 with the ability to degrade biofilm matrices.

Phylogenetic analysis of the tail fiber protein, a key determinant of host recognition often associated with depolymerase activity, was conducted using MEGA 11. As shown in Figure 9, P-07 shares the closest evolutionary relationship with *Proteus* phage P16-2532 (GenBank accession: QHJ72758), with a coverage of 71% and a high amino acid identity of 98.29%, strongly supporting their common classification within the genus *Lavrentievirus*. In contrast, P-07 is distantly related to *Gorganvirus* phage vB_PmiS_PM-CJR (GenBank accession: QYC51789), which is also reported to possess depolymerase activity. This disparity indicates that the genetic determinants for depolymerase production can be found across distinct phage genera with divergent evolutionary backgrounds (Figure 9).

## 4. Discussion

Antibiotic misuse has rendered multidrug-resistant (MDR) bacteria a global public health threat [23]. As an opportunistic pathogen, *P. mirabilis* causes diarrhea, vomiting, and urinary tract infections, with MDR strains posing severe challenges. Bacteriophages represent promising candidates for controlling MDR pathogens and biofilm infections. In this study, we isolated a lytic phage, designated vB_Pmc_P-07 (P-07), from fecal-wastewater samples, which belongs to the *Caudoviricetes* class. The phage formed plaques surrounded by distinct translucent halos—a phenotypic indicator of potential depolymerase production, which is a feature highly relevant for combating biofilm-associated infections. Guided by this observation, we conducted genomic and structural analyses that revealed a C-terminal domain (residues 178–225) in the tail fiber protein (ORF31) with structural homology to lyase enzymes, supporting its putative depolymerase function. This molecular evidence, combined with the halo phenotype, indicates that P-07 may encode enzymatic machinery capable of degrading bacterial exopolysaccharides—a crucial ability for penetrating and disrupting biofilms. However, this functional attribution remains speculative and warrants further validation through studies such as the heterologous expression of ORF31 and enzymatic assays using exopolysaccharide substrates.

Characterizing phage biology is essential for developing therapeutic strategies against zoonotic infections. P-07 exhibited a narrow host range, lysing only *P. mirabilis* without cross-species activity against *E. coli*, *K. pneumoniae*, or *S. aureus*. This finding is consistent with Pan et al. [24] and reflects receptor-binding specificity. Although its host range is limited, this phage enriches the *Proteus* phage repository and holds clinical potential. One-step growth analysis revealed advantageous reproductive replication kinetics: a latent period of 30 min and a burst size of 474.99 PFU/cell, outperforming phage PmP19 (40 min, 352 PFU/cell) [25]. The shorter latent period indicates a quicker onset of lytic activity, which is therapeutically beneficial for rapid bacterial clearance, while the larger burst size implies higher replicative efficiency, potentially leading to more robust and sustained antibacterial action. Furthermore, P-07 demonstrated notable environmental stability, retaining high viability at 40–60 °C and pH 4.0–12.0, which is comparable to the resilience reported for phage vB_PmM_S, vB_PmM_W, vB_PmM_X [26].

The formidable resistance of *P. mirabilis* biofilms, which can elevate antibiotic tolerance up to 1000-fold compared to planktonic cells [27], highlights the need for alternative therapeutic strategies. In this context, phage-based approaches have shown considerable promise. For instance, Mirzaei et al. [28] documented that a phage cocktail could effectively suppress *P. mirabilis* colonization on urinary catheters, while Soontarach et al. [29] reported concentration-dependent biofilm inhibition by phage LysAB1245 against *Pseudomonas aeruginosa* and *Staphylococcus aureus*. Consistent with these reports, our results confirm the anti-biofilm capacity of P-07 against porcine-originated MDR *P. mirabilis*, showing significant biomass reduction across multiple time points and phage concentrations. Removal efficacy increased over time, with the strongest effect at 6 h post-treatment—a pattern indicative of cumulative lytic and enzymatic degradation. These findings align with previous work by Esmael et al. [21] and support the potential of P-07 as a candidate for treating biofilm-associated infections.

Genomic analysis of P-07 (58,582 bp, 46.91% GC content, 63 ORFs) expands *Proteus* phage databases. Phylogenetically classified under *Casjensviridae* family and *Lavrentievirus* genus, it resembles phage vB_PmiP_pPm05, though with a shorter genome, and shares similarities with RS1pmA (42,112 bp, 41.10% GC) [8]. Notably, P-07 encodes only one lysis protein (ORF24) versus four in RS1pmA. Functional annotation categorized proteins into four modules: genes related to metabolism, genes related to lysis, structural and packaging genes, and genes related to DNA replication and regulation. A terminase complex ensured precise genome encapsidation [30]. Phylogenetic analysis of the terminase large subunit indicated that P-07 is most closely related to *P. mirabilis* phages PM87 and pPM01.

Tail fiber protein phylogeny offers evolutionary insights into the distribution of depolymerase function. While P-07 shares the closest relationship with *Proteus* phage P16-2532, it is distantly related to *Gorganvirus* phage vB_PmiS_PM-CJR, which also possesses depolymerase activity. This evolutionary divergence indicates independent acquisition of depolymerase capability across distinct phage genera, highlighting convergent evolution of this therapeutically valuable trait.

Critically, no virulence or antibiotic resistance genes were detected, substantially reducing risks in therapeutic development. Compared to traditional antibiotics, P-07 represents a safer clinical alternative without resistance concerns—a vital advancement against the MDR crisis.

## Figures and Tables

**Figure 1 viruses-17-01419-f001:**
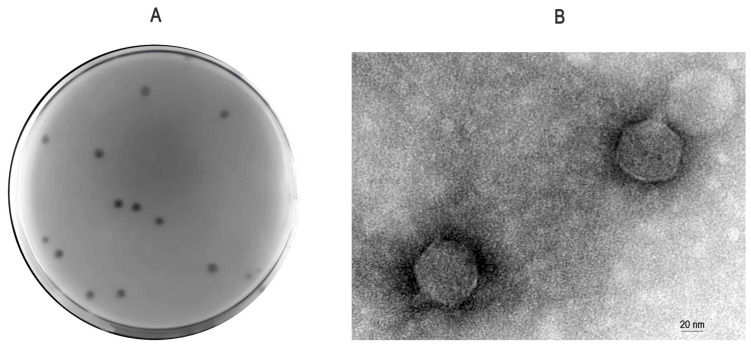
Morphological characteristics of bacteriophage vB_Pmc_P-07 (P-07). (**A**) Plaque morphology observed on a lawn of *Proteus mirabilis*. Clear plaques surrounded by translucent halos are visible; (**B**) Transmission electron micrograph of an individual P-07 virion negatively stained with 2% phosphotungstic acid (imaging performed at 80 kV). The phage particle exhibits an icosahedral head and a short tail. Scale bar, 20 nm.

**Figure 2 viruses-17-01419-f002:**
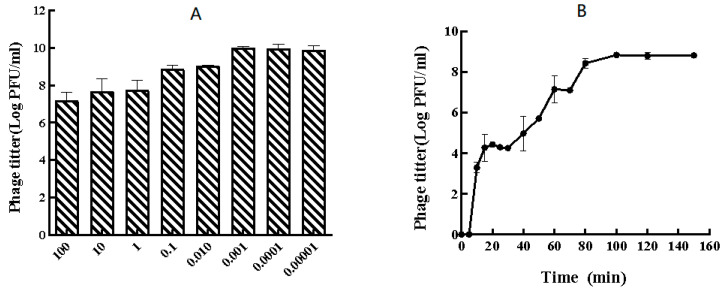
Replication characteristics of bacteriophage vB_Pmc_P-07 (P-07). (**A**) Determination of the optimal multiplicity of infection (MOI). The phage titer was measured after infection at different MOIs, with the highest yield observed at an MOI of 0.001; (**B**) One-step growth curve of P-07. The phage exhibited a latent period of approximately 30 min, followed by a rapid rise phase and a final plateau at 100 min, resulting in a burst size of 474.99 PFU/cell. Error bars represent the standard deviation from triplicate experiments.

**Figure 3 viruses-17-01419-f003:**
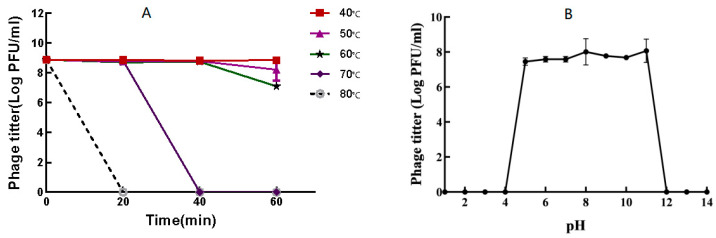
Environmental stability of bacteriophage vB_Pmc_P-07 (P-07). (**A**) Thermal stability. Phage viability was assessed after incubation at various temperatures (40–80 °C) for up to 60 min. P-07 maintained stability at 40–60 °C but was rapidly inactivated at 70 °C; (**B**) pH tolerance. Phage activity was evaluated after exposure to a broad pH range (pH 1–14) for 2 h. P-07 retained high infectivity within the pH 5–11 range, while being inactivated under strongly acidic (pH ≤ 4) or alkaline (pH ≥ 12) conditions.

**Figure 4 viruses-17-01419-f004:**
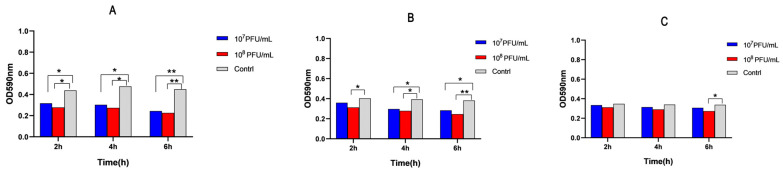
Concentration and time dependent anti-biofilm activity of bacteriophage P-07 against preformed *P*. *mirabilis* biofilms. (**A**) Anti-biofilm efficacy against 24 h mature biofilms formed by the strong biofilm-forming strain *P. mirabilis* 07. Biofilms were treated with low- (5.6 × 10^7^ PFU/mL) or high-concentration (5.6 × 10^8^ PFU/mL) phage suspensions for 2, 4, and 6 h. Both concentrations significantly reduced biofilm biomass compared to the control at all time points (* *p* < 0.05; ** *p* < 0.01); (**B**) Anti-biofilm efficacy against 24 h mature biofilms formed by the moderate biofilm-forming strain *P. mirabilis* 08. Significant reductions were observed at all time points, with enhanced inhibition at 6 h (* *p* < 0.05; ** *p* < 0.01); (C) Anti-biofilm efficacy against 24 h mature biofilms formed by the weak biofilm-forming strain *P. mirabilis* 13. Significant reduction was only observed in the high-concentration group after 6 h of treatment (* *p* < 0.05).

**Figure 5 viruses-17-01419-f005:**
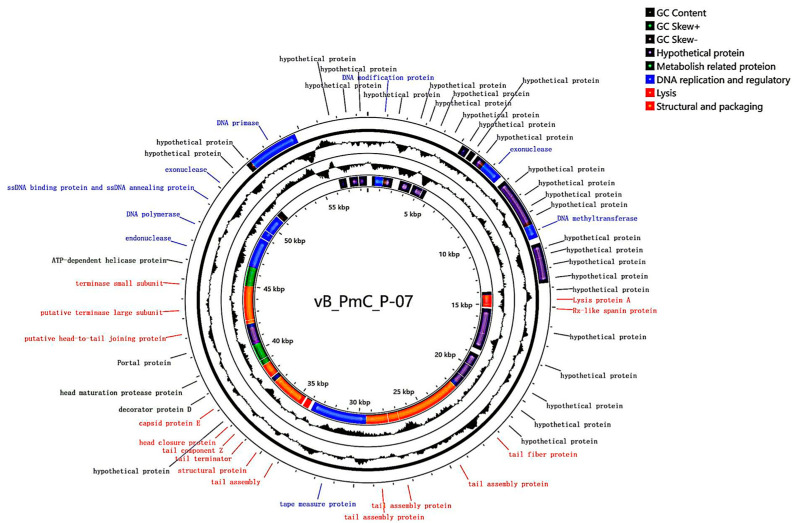
Circular genome map of bacteriophage vB_Pmc_P-07 (P-07). The map depicts the complete double-stranded DNA genome (58,582 bp). Open Reading Frames (ORFs) on the forward and reverse strands, color-coded by functional module (structural/packaging: orange; DNA replication/regulation: blue; lysis: red; auxiliary metabolic: green).

**Figure 6 viruses-17-01419-f006:**
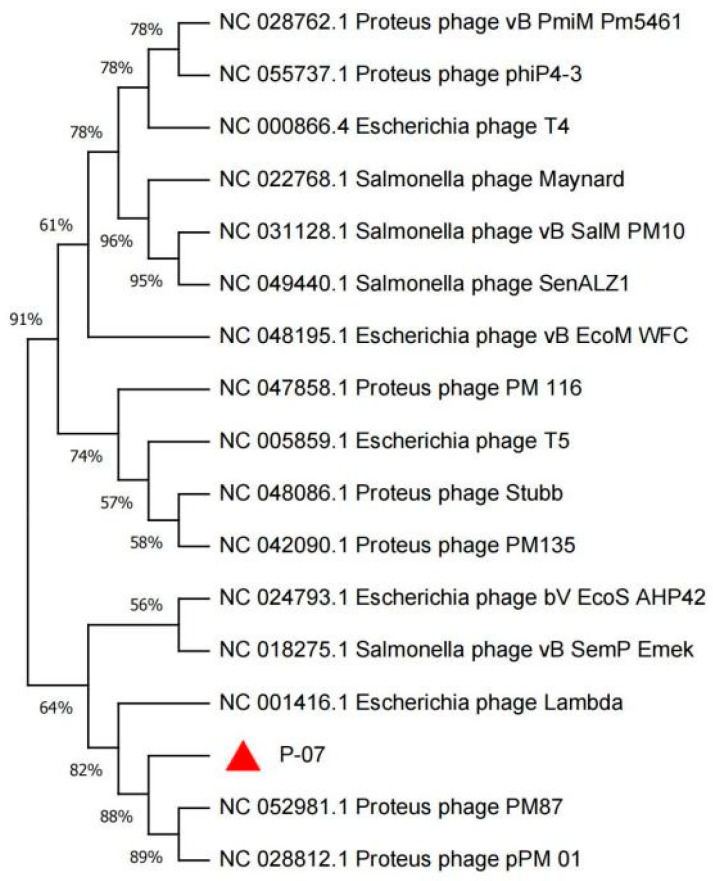
Phylogenetic analysis of the terminase large subunit of bacteriophage P-07. The maximum-likelihood tree was constructed based on amino acid sequence alignment, confirming the evolutionary position of P-07 (indicated by a red triangle) within the class *Caudoviricetes*, family *Casjensviridae*, and genus *Lavrentievirus*. Bootstrap values (≥70%) are shown at branch nodes.

**Figure 7 viruses-17-01419-f007:**
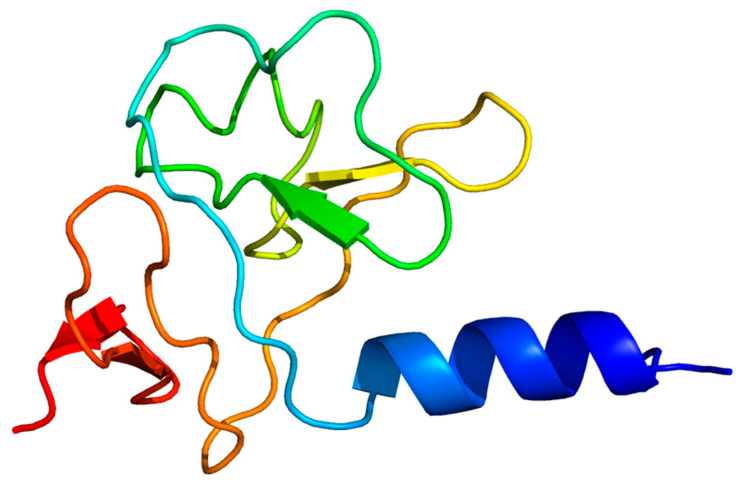
Tertiary structure prediction and domain architecture of phage P-07 tail fiber protein. The tertiary structure model of the ORF31 tail fiber protein is presented, coloured in a rainbow spectrum from the N-terminus (blue) to the C-terminus (red). Phyre2 modeling identified critical functional domains in ORF31, including a structural N-terminal region (residues 22–115; 38.817 × 37.000 × 30.111 Å) homologous to baseplate complexes and a C-terminal region with lyase homology. Secondary structure composition (9% α-helix, 33% β-sheet) supports its multifunctional role in phage infection.

**Figure 8 viruses-17-01419-f008:**
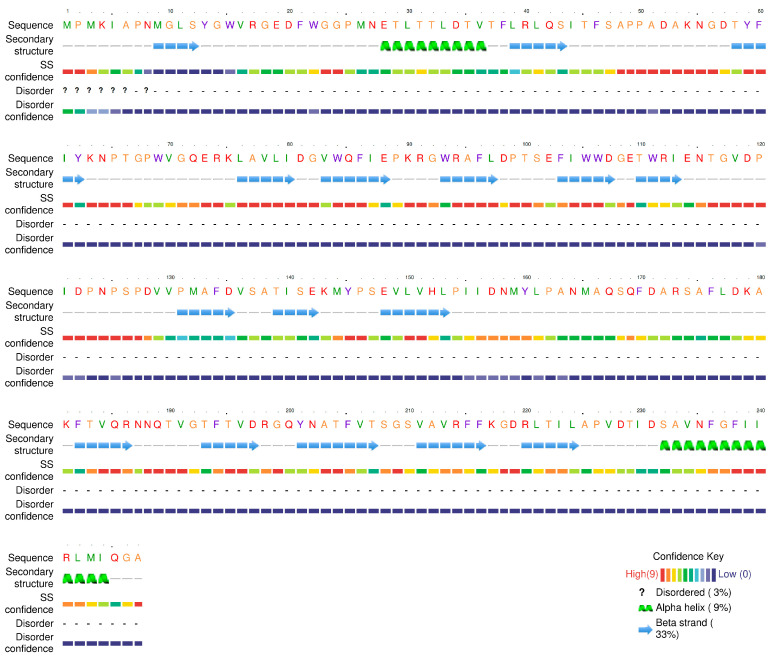
Secondary structure composition and functional implications of the tail fiber protein (ORF31). Analysis of the predicted structure reveals a distinct secondary structure profile dominated by β-sheets (33%) with minimal α-helical content (9%). This composition is consistent with the protein’s dual functional role, providing both structural stability for baseplate integration and flexibility for enzymatic activity in the C-terminal domain.

**Figure 9 viruses-17-01419-f009:**
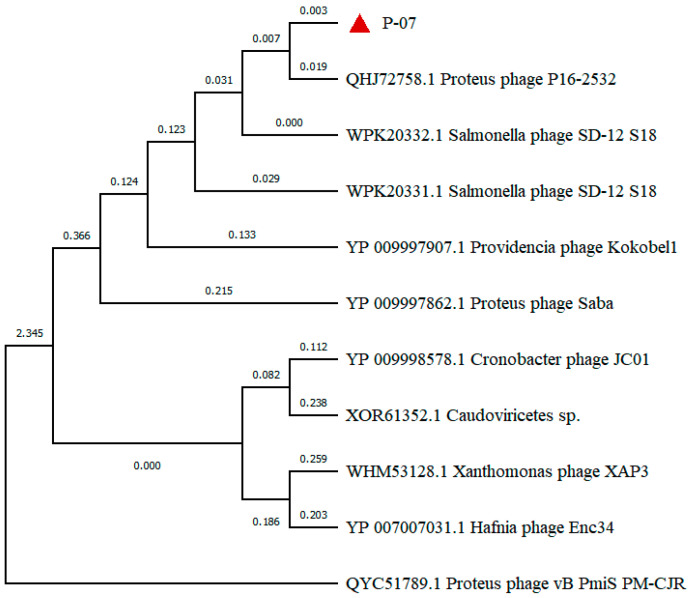
Phylogenetic analysis and evolutionary relationships of the tail fiber protein. The maximum-likelihood tree was constructed based on amino acid sequence alignment, illustrating the phylogenetic placement of P-07 ORF31 (indicated by a red triangle) among related phage tail fiber proteins.

**Table 1 viruses-17-01419-t001:** Genomic features and functional annotation of phage vB_Pmc_P-07.

ORF	START	STOP	Functional Module	Function	Identity/%	E Value	Accession Number
46	40,452	39,175	Metabolism-related protein	head maturation protease protein	99.76%	0	YP_009199670.1
51	46,549	45,026		ATP-dependent helicase protein	99.80%	0	UGO49911.1
2	1183	545	DNA replication and regulatory	DNA modification	99.53%	2.00 × 10^−153^	UGO51540.1
13	6597	7685		exonuclease	100.00%	0	YP_009997974.1
18	10,605	11,342		DNA methyltransferase	99.59%	0	UGO36038.1
37	33,756	29,425		tape measure	99.79%	0	UVD32353.1
52	46,796	46,509		endonuclease	78.02%	4.00 × 10^−47^	YP_009997885.1
53	48,829	46,796		DNA polymerase	99.11%	0	UNI72738.1
54	49,499	48,894		binding and annealing	98.99%	1.00 × 10^−139^	YP_009998014.1
55	50,859	49,531		exonuclease	97.74%	0	YP_010187223.1
58	52,039	54,714		DNA primase	99.78%	0	UNI72672.1
24	14,961	14,257	Lysis	lysis protein A	100.00%	1.00 × 10^−170^	UGO51513.1
25	15,286	14,972	Structural and packaging	Rz-like spanin	100.00%	4.00 × 10^−34^	YP_009199647.1
31	22,622	21,879		tail fiber	99.60%	2.00 × 10^−179^	YP_009199654.1
32	26,443	22,622		tail assembly	99.29%	0	UGO48774.1
33	26,645	26,436		tail assembly	81.54%	5.00 × 10^−31^	YP_009997909.1
34	26,881	26,645		tail assembly	82.05%	1.00 × 10^−39^	YP_009997910.1
35	27,709	26,894		tail assembly	82.29%	5.00 × 10^−171^	YP_009997911.1
36	29,419	27,731		tail assembly	99.29%	0	YP_009997997.1
38	34,449	33,997		tail assembly	100.00%	8.00 × 10^−102^	UGO51498.1
39	35,775	34,588		structural	99.75%	0	UGO49923.1
40	36,291	35,779		tail terminator	99.41%	9.00 × 10^−119^	YP_009199664.1
41	36,898	36,281		tail component Z	99.51%	1.00 × 10^−144^	WOZ54555.1
42	37,254	36,901		head closure	100.00%	2.00 × 10^−80^	UGO36011.1
44	38,732	37,665		capsid E	99.72%	0	UGO49918.1
45	39,161	38,745		decorator protein D	99.28%	2.00 × 10^−94^	YP_009199669.1
48	42,367	42,092		head-to-tail joining	97.80%	2.00 × 10^−59^	UGO51620.1
49	44,446	42,380		terminase large subunit	100.00%	0	UGO51619.1
50	45,023	44,439		terminase small subunit	99.48%	2.00 × 10^−141^	UXY92233.1
1	548	327	Hypothetical protein	hypothetical protein	98.63%	2.00 × 10^−46^	WP_196727143.1
3	1927	1187		hypothetical protein	99.59%	7.00 × 10^−177^	WOZ54581.1
4	3403	2834		hypothetical protein	99.47%	6.00 × 10^−135^	WOZ54583.1
5	3910	3509		hypothetical protein	100.00%	2.00 × 10^−94^	QHJ72730.1
6	4079	3891		hypothetical protein	100.00%	3.00 × 10^−38^	UGO49958.1
7	4311	4069		hypothetical protein	97.50%	2.00 × 10^−47^	EKV7661800.1
8	4649	4323		hypothetical protein	100.00%	5.00 × 10^−72^	YP_009199627.1
9	5119	5550		hypothetical protein	99.30%	4.00 × 10^−97^	WP_407818269.1
10	5632	5943		hypothetical protein	100.00%	9.00 × 10^−68^	WP_318768141.1
11	5949	6269		hypothetical protein	100.00%	7.00 × 10^−72^	WP_196727151.1
12	6263	6610		hypothetical protein	100.00%	3.00 × 10^−75^	WP_196727152.1
14	7955	8773		hypothetical protein	100.00%	0	WOZ54596.1
15	8770	9537		hypothetical protein	99.61%	0	WOZ54597.1
16	9534	9980		hypothetical protein	100.00%	8.00 × 10^−107^	YP_009998137.1
17	9973	10,605		hypothetical protein	99.52%	3.00 × 10^−153^	WOZ54599.1
19	11,644	12,177		hypothetical protein	100.00%	4.00 × 10^−129^	WP_196727159.1
20	12,170	12,421		hypothetical protein	97.59%	3.00 × 10^−52^	UNI72704.1
21	12,423	13,292		hypothetical protein	100.00%	0	WP_196727160.1
22	13,289	13,753		hypothetical protein	97.40%	1.00 × 10^−108^	EKV7661814.1
23	14,257	14,003		hypothetical protein	98.81%	2.00 × 10^−51^	YP_009199645.1
26	17,282	15,297		hypothetical protein	96.52%	0	WP_318768180.1
27	18,926	17,286		hypothetical protein	91.74%	0	WOZ54610.1
28	19,952	18,936		hypothetical protein	99.41%	0	YP_009199651.1
29	20,948	19,965		hypothetical protein	100.00%	0	YP_009199652.1
30	21,879	20,959		hypothetical protein	99.35%	0	UNI72715.1
43	37,601	37,254		hypothetical protein	98.26%	3.00 × 10^−73^	UXY92226.1
56	51,402	50,920		hypothetical protein	96.88%	1.00 × 10^−103^	UGO51612.1
57	51,755	52,039		hypothetical protein	79.79%	2.00 × 10^−48^	YP_009997934.1
59	56,362	56,123		hypothetical protein	97.47%	2.00 × 10^−45^	UGO51678.1
60	56,937	56,359		hypothetical protein	98.44%	9.00 × 10^−140^	YP_009199616.1
61	57,146	56,934		hypothetical protein	98.57%	4.00 × 10^−41^	UGO49967.1
62	57,848	57,159		hypothetical protein	97.38%	9.00 × 10^−164^	WOZ54577.1
63	58,509	57,901		hypothetical protein	93.56%	4.00 × 10^−136^	UGO51542.1

## Data Availability

Data supporting the findings are presented in the article.

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
