# Peer review of "Genomic and Biological Characterization of a Novel Proteus mirabilis Phage with Anti-Biofilm Activity"

_viruses, 2025, doi:10.3390/v17111419_

Round 1

Reviewer 1 Report

Comments and Suggestions for Authors

This manuscript by Liang et al. reports the isolation and thorough characterization of a novel lytic bacteriophage, vB_Pmc_P-07, which specifically targets a multidrug-resistant strain of Proteus mirabilis. The study is well-organized, methodologically robust, and provides a comprehensive dataset encompassing the phage’s biological properties, stability profile, anti-biofilm efficacy, and genomic characteristics. These findings represent a valuable addition to the field of phage therapy, particularly in addressing biofilm-associated infections within veterinary and public health settings. The manuscript is clearly written and merits publication after minor revisions.

1.There is an inconsistency in the numbering of sections within the Results. Section 3.7 ("Biofilm Clearance Efficacy") should be renumbered to restore sequential order (likely as Section 3.5).

2.Please specify the accelerating voltage used during transmission electron microscopy (TEM). Including this parameter in both the Results section and the figure legend would improve reproducibility and scientific rigor.

3.Table 1 contains several typographical errors—e.g., “Metabolish related proteion” should be corrected to “Metabolism-related protein,” and “Lavrentievirus” is misspelled. A careful review of the table and text is recommended.

4.In Figure 1, the species name “Proteus mirabilis” in the caption for panel A should be italicized.

5.Although indicating “data not yet published” for the antibiotic resistance profile is acceptable, briefly describing the antimicrobial susceptibility testing method (e.g., disk diffusion or MIC assay) would be preferable.

6.The discussion regarding depolymerase activity is compelling but remains speculative. It would be appropriate to explicitly indicate that future studies—such as heterologous expression and enzymatic assays of ORF31—are necessary to confirm this function.

7.When comparing burst size and latent period to those of other phages (e.g., PmP19), please include a brief commentary on how a shorter latent period and larger burst size may offer therapeutic advantages.

Author Response

Comments 1: There is an inconsistency in the numbering of sections within the Results. Section 3.7 ("Biofilm Clearance Efficacy") should be renumbered to restore sequential order (likely as Section 3.5).

Response 1: We sincerely thank the reviewer for this careful observation. The numbering inconsistency in the Results section has now been corrected.

1.Section 3.7 "Biofilm Clearance Efficacy Against Proteus mirabilis" has been renumbered as Section 3.5 to restore the correct sequential order (Line 275).

2.The original Section 3.8 "Whole-Genome Analysis of Phage P-07" is now correctly numbered as Section 3.6.(Line 294).

Comments 2: Please specify the accelerating voltage used during transmission electron microscopy (TEM). Including this parameter in both the Results section and the figure legend would improve reproducibility and scientific rigor.

Response 2: We thank the reviewer for this valuable suggestion.

The accelerating voltage used for TEM observation has now been specified in both Results section (Line 227) and the Figure 1 legend (Line 234) to enhance methodological reproducibility.

Comments 3: Table 1 contains several typographical errors—e.g., “Metabolish related proteion” should be corrected to “Metabolism-related protein,” and “Lavrentievirus” is misspelled. A careful review of the table and text is recommended.

Response3: We sincerely thank the reviewer for pointing out these typographical errors. We have carefully reviewed and corrected all instances in Table 1 and throughout the manuscript:

1."Metabolish related proteion" has been corrected to "Metabolism-related protein"(Table 1, Line 424).

2."Lavrentievavirus" has been corrected to "Lavrentievirus" (the correct spelling)(Line 324,332,363,421)

We have conducted a thorough proofreading of the entire table and manuscript text to ensure terminology consistency and accuracy. These corrections have been implemented in the revised version of the manuscript. We appreciate the reviewer's attention to detail in helping us improve the quality of our presentation.

Comments 4: In Figure 1, the species name “Proteus mirabilis” in the caption for panel A should be italicized.

Response 4: We thank the reviewer for highlighting this oversight. The species name "Proteus mirabilis" in the caption for Figure 1A has now been italicized as required by standard microbiological nomenclature (Line 232).

Comments 5:Although indicating “data not yet published” for the antibiotic resistance profile is acceptable, briefly describing the antimicrobial susceptibility testing method (e.g., disk diffusion or MIC assay) would be preferable.

Response 5: We thank the reviewer for this constructive suggestion. We have now included the methodological details of antimicrobial susceptibility testing in the revised manuscript.

1.The text in Section 2.1 has been updated as follows:

"Pm 07 is an ESBL-producing P. mirabilis strain that exhibits resistance to 18 antibiotics... as determined by the Kirby-Bauer disk diffusion method on Mueller-Hinton agar, with results interpreted according to the Clinical and Laboratory Standards Institute (CLSI) guidelines" (Line66-68).

2.Additionally, we have supplemented the description of biofilm formation capability analysis in the same section:

"The biofilm-forming ability of strain Pm 07 was evaluated using the crystal violet staining method, demonstrating strong biofilm formation with an OD₅₉₀ value exceeding 4-fold that of the negative control (data not yet published)."(Line 68-70).

Comments 6: The discussion regarding depolymerase activity is compelling but remains speculative. It would be appropriate to explicitly indicate that future studies—such as heterologous expression and enzymatic assays of ORF31—are necessary to confirm this function

Response 6: We appreciate the reviewer's insightful comment regarding the discussion of depolymerase activity. In response to this suggestion, we have revised the Discussion section to explicitly state the speculative nature of this functional attribution and to outline the necessary future studies for verification.

The relevant paragraph in the Discussion section now reads:

"This molecular evidence, combined with the halo phenotype, indicates that P-07 may encode enzymatic machinery capable of degrading bacterial exopolysaccharides—a crucial ability for penetrating and disrupting biofilms. However, this functional attribution remains speculative and warrants further validation through studies such as the heterologous expression of ORF31 and enzymatic assays using exopolysaccharide substrates."(Line 386-390).

Comments 7: When comparing burst size and latent period to those of other phages (e.g., PmP19), please include a brief commentary on how a shorter latent period and larger burst size may offer therapeutic advantages..

Response 7: We thank the reviewer for this valuable suggestion. We have now added a commentary on the therapeutic implications of the phage's reproductive parameters in the Discussion section.

The revised text reads:

"One-step growth analysis revealed advantageous reproductive replication kinetics: a latent period of 30 minutes and a burst size of  474.99 PFU/cell, outperforming phage PmP19 (40 minutes, 352 PFU/cell) [23]. The shorter latent period indicates a quicker onset of lytic activity, which is therapeutically beneficial for rapid bacterial clearance, while the larger burst size implies higher replicative efficiency, potentially leading to more robust and sustained antibacterial action. "(Line 398-401).

Reviewer 2 Report

Comments and Suggestions for Authors

The study characterizes the genome and biological properties of a new bacteriophage P-07, targeting MDR Proteus mirabilis strain, with particular interest in its potential anti-biofilm activity.

The manuscript is well written and the data are presented clearly. The results are
described logically and in a structured manner, that renders their meaning clear and easily understandable. However, some minor points need to be addressed.

Introduction: The ultimate aim of the study is supposed to be the application of the novel bacteriophage in swine farming, so the introduction should focus more on this field. What is the impact of zoonotic infection by P. mirabilis on animals in terms of symptoms, livestock mortality and management costs? In addition, how could bacteriophages be applied specifically in livestock farming? There is already data in the literature regarding the use of bacteriophages in various types of animal husbandry as a treatment for E. coli and Salmonella infections (doi: 10.3390/ani15121713; doi: 10.1016/j.psj.2025.105595). Is there any data on their effectiveness against P. mirabilis? Please improve the introductory paragraph with this background information, as it would help to highlight the significance of the study. 

Line 39-40: delete the repeated sentence 

Line 57: where was the strain isolated from?

Line 68: define the acronym for “SM”

Line 80: delete the repeated "was"

Line 74-90: uniform the units (rpm orLine 153-162-1 r/min) throughout the text 

Line 100: uniform with "×"

Line 108: what volume was used to resuspend the pellet?

Line 153-162-181: uniform with "standard deviation"

Line 208: please add what was considered a statistically significant P value (e.g. p≤ 0.05)

Line 221: correct P. mirabilis in italics 

Figure 1: is the scale bar correct? From the TEM image, the head diameter of the virions (60 nm) appears longer than the scale (200nm)

Figure 3A: please add how the phage titre drops at 80°, shown in the image but not mentioned in the text. 

Figure 5: please increase the resolution of the text to improve readability

Table 1: correct "STAR" and add the meaning of "E" data

Line 361-364: combine the two sentences with a repeated notion 

Author Response

Comments 1: Introduction: The ultimate aim of the study is supposed to be the application of the novel bacteriophage in swine farming, so the introduction should focus more on this field. What is the impact of zoonotic infection by P. mirabilis on animals in terms of symptoms, livestock mortality and management costs? In addition, how could bacteriophages be applied specifically in livestock farming? There is already data in the literature regarding the use of bacteriophages in various types of animal husbandry as a treatment for E. coli and Salmonella infections (doi: 10.3390/ani15121713; doi: 10.1016/j.psj.2025.105595). Is there any data on their effectiveness against P. mirabilis? Please improve the introductory paragraph with this background information, as it would help to highlight the significance of the study.

Response 1: We thank the reviewer for these insightful suggestions. We have thoroughly revised the Introduction to better highlight the application potential in swine farming and address the specific points raised (Line 32-54). The main improvements include:

1.We have streamlined the Introduction by removing less relevant content.

2.Enhanced focus on swine farming impacts: We have added discussion of how P. mirabilis affects swine health and production, and its role as a reservoir for antimicrobial resistance.

3. Specific application contexts for phage therapy: We have incorporated examples of how bacteriophages could be practically deployed in livestock operations, citing the suggested references on phage applications against E. coli and Salmonella (doi: 10.3390/ani15121713; doi: 10.1016/j.psj.2025.105595). The references have been correspondingly labeled as [10], [11] (Line 484-488).

Comments 2: Line 39-40: delete the repeated sentence

Response 2: We thank the reviewer for pointing out this issue. As the Introduction has been systematically revised and restructured during the revision process, the repetitive sentence mentioned has been naturally eliminated in the current version.

We have verified the entire manuscript to ensure clarity and absence of repetition.

Comments 3: Line 57: where was the strain isolated from?

Response3: We thank the reviewer for this question. The host strain P. mirabilis Pm 07 and other tested strains (41 strains of P. mirabilis, 18 strains of E. coli, 3 strains of Klebsiella pneumoniae, and two strains of Staphylococcus aureus) were isolated from Xinjiang Uygur Autonomous Region of China. The sources of all bacterial strains have been supplemented in the revised manuscript.(Line 59).

Comments 4: Line 68: define the acronym for “SM”

Response 4: We thank the reviewer for this suggestion. The acronym "SM" has now been defined at its first occurrence in the text.

The revised sentence in Section 2.2 reads:

"Briefly, SM buffer ( 50 mM Tris-HCl [pH 7.5], 100 mM NaCl, 8 mM MgSOâ‚„, and 0.01% gelatin) was added to mixed fecal-sewage samples..."(Line 75-76).

Comments 5:Line 80: delete the repeated "was"

Response 5: We thank the reviewer for pointing out this grammatical error. The repeated "was" in Line 80 (now Line 88 in the revised manuscript) has been deleted.

The corrected sentence now reads:

"Briefly, 200 μL log period P. mirabilis culture was spread evenly onto LB solid medium using a sterile cotton swab."  (Line88).

Comments 6: Line 74-90: uniform the units (rpm orLine 153-162-1 r/min) throughout the text

Response 6: We thank the reviewer for pointing out this inconsistency. We have standardized the rotational speed units throughout the manuscript by replacing all "rpm" instances with "r/min" to maintain consistency with SI unit conventions.

The corrections include:

Line 82: "180 r/min" (previously "rpm")

Line 181: "180 r/min" (previously "rpm")

Comments 7: Line 100: uniform with "×"

Response 7: We thank the reviewer for pointing out this inconsistency. We have standardized the multiplication sign throughout the manuscript by replacing all similar instances with the proper "×" symbol.

The specific correction in Line 108 has been made:

 “The supernatant was collected by centrifugation (7,450 × g, 15 min, 4℃ )”.

We have verified that all centrifugation parameters and other numerical values involving multiplication now consistently use the " ×" symbol throughout the manuscript.

Comments 8: Line 108: what volume was used to resuspend the pellet?

Response8: We thank the reviewer for raising this important methodological detail. We have now specified the resuspension volume in the relevant procedure description.

The text in Section 2.4 has been revised to read:

"The resulting pellet was resuspended in 1 mL of SM buffer and incubated at 4 ℃ for 1 h." (Line 116).

Additionally, we have replaced the vague reference to "a large-volume phage lysate" with the specific volume used:"Briefly, a 200 mL phage lysate was initially prepared." (Line 107).

Comments 9: Line 153-162-181: uniform with "standard deviation"

Response9: We thank the reviewer for this suggestion. We have now standardized the terminology for statistical variation throughout the manuscript by using "mean ± SD." consistently.

The inconsistency in Line 190 has been corrected.

Comments 10: Line 208: please add what was considered a statistically significant P value (e.g. p≤ 0.05)

Response10: We thank the reviewer for this suggestion. We have now specified the thresholds for statistical significance in the Methods section.

The text in Section 2.12 has been updated as follows:

"Significance was determined by multiple t-tests. Differences were considered statistically significant at P < 0.05 and highly significant at P < 0.01. " (Line 218-219).

Comments 11: Line 221: correct P. mirabilis in italics

Response11: We thank the reviewer for highlighting this oversight. The species name "Proteus mirabilis" in the caption for Figure 1A has now been italicized as required by standard microbiological nomenclature (Line 232).

Comments 12: Figure 1: is the scale bar correct? From the TEM image, the head diameter of the virions (60 nm) appears longer than the scale (200nm)

Response12: We sincerely thank the reviewer for this careful observation. The scale bar in the original Figure 1B was indeed incorrect. We have now replaced the TEM image with a new micrograph that includes an accurate 20 nm scale bar, which correctly corresponds to the indicated virion dimensions (approximately 60 nm in head diameter).

The Figure 1 legend has been updated accordingly:

"(B) Transmission electron micrograph of an individual P-07 virion negatively stained with 2% phosphotungstic acid ( imaging performed at 80 kV). The phage particle exhibits an icosahedral head and a short tail. Scale bar, 20 nm." (Line 235)

We apologize for this oversight and confirm that all morphological descriptions in the text are consistent with the scaled measurements from the corrected image.

Comments 13: Figure 3A: please add how the phage titre drops at 80°, shown in the image but not mentioned in the text.

Response13: We thank the reviewer for this valuable observation. We have now supplemented the description of phage stability at 80°C in the Results section (Section 3.4) to align with the data shown in Figure 3A.

The revised text now reads:

"However, at 70 ℃, the phage titer declined rapidly after 20 minutes and was completely inactivated by 40 minutes. At 80 ℃, the phage was completely inactivated after 20 minutes of incubation, with no plaques detected." (Line 261-262).

Comments 14: Figure 5: please increase the resolution of the text to improve readability

Response14: We thank the reviewer for this suggestion. We have replaced Figure 5 with a new high-resolution image in which all textual elements have been enhanced for optimal readability. The updated figure now clearly displays all genomic features and annotations.

Comments 15: Table 1: correct "STAR" and add the meaning of "E" data

Response15: We thank the reviewer for pointing out these important issues in Table 1. We have made the following corrections:

The column header "STAR" has been corrected to "START" (indicating the start position of each ORF).

The column header "E" has been updated to "E-value" to clearly indicate that these values represent the Expect value from BLAST analysis, reflecting the statistical significance of sequence matches.

We have carefully reviewed the entire table to ensure all terminology is accurate and consistently presented.

Comments 16: Line 361-364: combine the two sentences with a repeated notion

Response16: We thank the reviewer for this suggestion. We have combined the two repetitive sentences into a single, more concise statement as follows:

Revised text:

"The phage formed plaques surrounded by distinct translucent halos—a phenotypic indicator of potential depolymerase production, which is a feature highly relevant for combating biofilm-associated infections." (Line 379-382).